# ALD Deposited ZnO:Al Films on Mica for Flexible PDLC Devices

**DOI:** 10.3390/nano11041011

**Published:** 2021-04-15

**Authors:** Dimitre Z. Dimitrov, Zih Fan Chen, Vera Marinova, Dimitrina Petrova, Chih Yao Ho, Blagovest Napoleonov, Blagoy Blagoev, Velichka Strijkova, Ken Yuh Hsu, Shiuan Huei Lin, Jenh-Yih Juang

**Affiliations:** 1Institute of Solid State Physics, Bulgarian Academy of Sciences, 1784 Sofia, Bulgaria; blago_sb@yahoo.com; 2Institute of Optical Materials and Technologies, Bulgarian Academy of Sciences, 1113 Sofia, Bulgaria; d_kerina@abv.bg (D.P.); blgv@abv.bg (B.N.); vily@iomt.bas.bg (V.S.); 3Department of Electrophysics, National Yang Ming Chiao Tung University, Hsinchu 30010, Taiwan; zifan.ep09g@nctu.edu.tw (Z.F.C.); ro3637678.py07g@nctu.edu.tw (C.Y.H.); lin@cc.nctu.edu.tw (S.H.L.); jyjuang@nctu.edu.tw (J.-Y.J.); 4Faculty of Engineering, South-West University “Neofit Rilski”, 2700 Blagoevgrad, Bulgaria; 5Photonics Department, National Yang Ming Chiao Tung University, Hsinchu 30010, Taiwan; ken@nctu.edu.tw

**Keywords:** Al-doped ZnO, ALD technique, transparent conductive layers, PDLC structures, flexible devices

## Abstract

In this work, highly conductive Al-doped ZnO (AZO) films are deposited on transparent and flexible muscovite mica substrates by using the atomic layer deposition (ALD) technique. AZO-mica structures possess high optical transmittance at visible and near-infrared spectral range and retain low electric resistivity, even after continuous bending of up to 800 cycles. Structure performances after bending tests have been supported by atomic force microscopy (AFM) analysis. Based on performed optical and electrical characterizations AZO films on mica are implemented as transparent conductive electrodes in flexible polymer dispersed liquid crystal (PDLC) devices. The measured electro-optical characteristics and response time of the proposed devices reveal the higher potential of AZO-mica for future ITO-free flexible optoelectronic applications.

## 1. Introduction

Recent technological appeal for flexible, lightweight, inexpensive, enabling low-power consumption, and environmentally-friendly optoelectronics has stimulated intensive research on new generation optoelectronic devices. In this aspect, flexible transparent electrodes attracted extreme attention in numerous applications to mention among others touch screens, flat panel displays, organic light-emitting diodes (OLEDs), flexible sensors, solar cells, electronic skins, etc. [1,2,3]. Currently, indium-tin oxide (ITO) is the material of choice for transparent conducting electrodes, however, because of the several limitations like indium scarcity in nature, increasing cost of the production processing, and the brittleness due to the ITO’s ceramic structure, the need of alternative transparent and conductive layers to replace ITO is critically important. As the main requirements for flexible electrodes are high optical transparency, low electrical resistance, and mechanical bending stability without a substantial drop in electro-optical performance, it is essential to reach an optimized balance between the sheet resistance, optical transmittance, and bending ability, e.g., to realize precise interface layer-substrate.

The wide band-gap semiconductor zinc oxide possesses plentiful of useful properties that have attracted increasing attention in copious fields of research. The high transparency in visible spectra of ZnO combined with its tunable electrical conductivity enables its use in applications ranging from thin film transistors (TFTs) to functional layers in photovoltaics. ZnO can be used in optoelectronic applications in the near UV spectral range, including light-emitting diodes (LEDs) and photodetectors due to its direct band gap of 3.37 eV. Piezoelectric properties of ZnO allow applications in sensors and micro-electromechanical systems. Most of these applications are made possible by a rich compositional and functional diversity achievable by doping of ZnO with various elements. ZnO and doped ZnO can be prepared as thin films using a variety of techniques. The preparation methods, properties, and applications are recently reviewed in [4,5]. However, the increasingly rigorous requirements of the microelectronics industry, among other factors, have led to a sharp increase in the use of the atomic layer deposition (ALD) technique in various thin-film applications.

So far, in terms of transparent conductive layers, several promising materials for instance transparent conductive oxides (TCOs), conductive polymers, graphene, carbon nanotubes (CNTs), etc., have been used as transparent electrodes possessing low resistance and high flexibility [6,7,8,9]. Among them, Al-doped zinc oxide (AZO) is mostly studied due to the high thermal stability, good resistance to hydrogen plasma damage, and lower fabrication cost, in comparison to the commercially available ITO layers [10]. In addition to similar electrical and optical properties to those of ITO, AZO has further advantages over ITO and some of the other common TCOs, including larger abundance, layers flexibility, and lack of toxicity [11]. Generally, AZO layers can be deposited by various methods as magnetron sputtering, sol-gel, pulsed laser deposition, metal-organic chemical vapor deposition at temperature interval between room temperature and 300 °C [12]. ALD technique is a deposition method relying on self-limiting reactions of specific precursors, characterized by exposing the substrate and growing layers to alternating organometallic precursors resulting in the sequential deposition of atomic layers onto the substrate surface [13]. The nature of the ALD reaction mechanism makes certain a high- quality pinhole-free coatings with an atomic-scale precision thickness control and excellent conformality onto large area arbitrary substrates with complex surface shapes, including 3D objects [14].

The current leading polymer materials for flexible optoelectronics, for example, polyethylene terephthalate (PET), polyethylene naphthalate (PEN), and polyimide (PI) show several limitations concerning the processing temperature, inferior dimensional stability, and frequently substantial differences of the thermal expansion coefficients between the polymer substrate and conductive electrode layer [15]. The common drawback of the above organic materials is that they cannot tolerate elevated temperatures (typically higher than 200 °C) and suffer from degradation over time. In contrast, it is well-known that mica, a natural transparent inorganic crystalline material, possesses high flexibility due to its sheet/layered structure. As a result of the weak van der Waals interactions between the adjacent layers (weak interlayer bonding), mica is easy to peel off resulting in facile cleavage along the {001} planes [16], elastic (Young’s modulus of 200 GPa), and easily bendable that allows for a variety of flexible applications [17,18,19]. Moreover, mica possesses excellent optical transmittance in the ultraviolet–visible-infrared range, high-temperature stability (up to 600 °C for muscovite mica), high dimensional stability, and low cost [20]. In addition, mica is chemically inert, non-toxic, and more importantly, is compatible with almost all deposition techniques. Based on all of the above, mica appears to be an ideal substrate for flexible devices and wearable optoelectronic applications [21].

Across such applications, polymer dispersed liquid crystal-based structures have recently attracted considerable attention for applications as outdoor displays, projection displays, switchable privacy glasses, energy-saving windows, light shutters, etc. [22,23]. Usually, typical polymer dispersed liquid crystal (PDLC)-based devices use conventional transparent ITO layers as electrodes; however, due to the brittleness of the ITO, its implementation in flexible optoelectronics is complicated.

Aluminum-doped ZnO films were grown by atomic layer deposition on different polymer flexible substrates such as PET, PEN, polyimide (Kapton) [24,25,26,27,28]. However, to the best of our knowledge, there is no report on the ALD of AZO layers on flexible mica substrates.

Herein, we report the performances of AZO films synthesized by the ALD technique on mica substrates and applications in flexible polymer dispersed liquid crystal (PDLC) structures. After deposition of AZO films on transparent mica, highly conductive and extremely foldable AZO electrodes were obtained, which displayed high optical transmittance and stable sheet resistance even after 800 cycles of bending. Sheet resistance was confirmed by the 4-point probe method and AFM analyses of the structure were conducted before and after bending tests. Based on performed characteristics, several flexible PDLC structures have been assembled and the measured electro-optical characteristics demonstrated great potential for future flexible optoelectronic applications.

## 2. Materials and Methods

AZO films were prepared by ALD using the Beneq TFS-200 system ((Beneq Group, Espoo, Finland)) on 34 µm-thick mica substrates at a deposition temperature of 200 °C. In addition, AZO films were deposited on 75 µm thick polyethylene terephthalate (PET) substrates for reference. The deposition temperature for AZO on PET was 100 °C. A buffer layer of Al_2_O_3_ (~15 nm) was preliminary deposited, at the same temperature, by ALD to prevent the interdiffusion between PET and AZO. In the ALD process diethylzinc (DEZ, Zn(C_2_H_5_)_2_), trimethylaluminum (TMA, Al(CH_3_)_3_), and deionized water (H_2_O) were used as precursors for Zn, Al, and oxidant, respectively. Pure nitrogen was used as a carrier and purge gas at an average flow of 600 sccm. The Al-doping concentration in ZnO was adjusted by varying the number of DEZ/H_2_O and TMA/H_2_O cycles in a standard procedure for ALD processes [29] as follows: after 24 cycles of DEZ/H_2_O, a cycle of TMA/H_2_O was applied consisting of one so-called ‘supercycle’. The pulse durations for precursors reactions were the same of 200 ms for DEZ, TMA, and H_2_O, whereas the purging time after each precursor reaction was 2 s. The substrates were fixed with heat-resistant tape (specified temperature range −75 to +260 °C) in the ALD reaction chamber.

The AZO film thicknesses were obtained by ellipsometric measurements of AZO on Si reference substrates and simulations using an M-2000D spectroscopic ellipsometer (J.A. Woollam and Co., Lincoln, NE, USA). The film thickness and the optical constants were determined by fitting the experimental Ψ and Δ data [30].

The experimental Ψ and Δ data were analyzed using a three-layer model consisting of a Si substrate with SiO_2_ native oxide as a first layer, a ZnO layer as a second layer, and a roughness layer as a third layer. For the Si substrate and the native oxide, the data from the database of CompleteEASE Woollam data analysis software was used. The AZO layer was modeled using a PSemi-M0 and two Gaussian oscillators. The roughness layer for all samples is modeled by Bruggeman’s EMA (effective medium approximation). The thickness of AZO on mica was estimated to approximately 173 nm, while the thickness of AZO on PET was about 150 nm, respectively.

Crystal structures of AZO-Mica were investigated by powder X-ray diffraction. The diffraction patterns were collected on a Bruker D8 Advance diffractometer with Cu Kα radiation (1.54056A) and LynxEye detector (Bruker Corporation, Billerica, MA, USA) within the 2Θ range from 5.3 to 80° and a constant step 0.02° 2Θ. The crystalline phase identification was performed with the Diffract-plus EVA using the ICDD-PDF2 Database. The surface topography and roughness of the as-grown films were examined by atomic force microscope (AFM) MFP-3D (Asylum Research, Oxford Instruments, NanoAnalysis 25.2 mi, High Wycombe, UK). A scanning electron microscope (SEM Philips 515, Koninklijke Philips N.V., Eindhoven, The Netherlands) was used to observe the morphologies of the prepared samples. The above structural measurements were performed also on AZO on PET for comparison.

The optical transmittance spectra were measured using an ultraviolet-visible-near-infrared (Shimadzu UV-VIS-NIR, Shimadzu Corporation, Kyoto, Japan) spectrophotometer, at room temperature.

The bending test was performed using a computerized home-built bending setup, with an ESP301 control platform. In the bending test experiments, the transparent AZO-mica electrode edges were fixed between two clamps horizontally and the samples were bent by pushing the two clamps together up to a bending radius of 1.5 mm. The sheet resistance was measured using the four-point probe technique (Ossila Ltd., Sheffield, UK, system) before and after the bending test. The electro-optical properties of AZO on mica were further assessed before and after bending tests to evaluate the flexibility performance.

Based on the above characteristics, several AZO-mica substrates were selected for PDLC device assembly. First, the PDLC mixture was made by mixing UV-curable monomer NOA65 (Norland, *n* = 1.524, Norland Products, Cranbury, NJ, USA) and E7 nematic liquid crystal (LC, Merck, Merck KGaA, Darmstadt, Germany) at a 30:70 wt.% ratio, using polymerization-induced phase separation method. The LC/monomer mixture was injected into an empty cell assembled by two AZO-mica substrates separated with a mylar spacer (12 µm thick). Next, the cells were exposed to an ultraviolet light source (365 nm) with an intensity of 60 Mw/cm^2^ for 15 min to polymerize the UV-curable monomer NOA65. As a result, the phase separation leads to the formation of randomly dispersed LC droplets in a polymer matrix, which determines the unique behavior of PDLC devices, as will be discussed later.

The electro-optical modulation of assembled AZO on mica PDLC devices was measured by placing each one between a pair of apertures in optical set-up. A helium-neon laser (He-Ne) emitting at 633 nm was used as a light source. The change of transmittance was monitored by the photodetector placed behind the aperture as a function of the applied root-mean-square (RMS) alternating current (AC) voltage, at 1 kHz frequency. To measure the response and fall time, a rectangular pulse with a certain voltage in the working range (parameters discussed in Section 3.4) of the device was applied. The characteristic response and decay curves were recorded with a digital storage oscilloscope.

## 3. Results

### 3.1. Structural and Morphological Properties

X-ray diffraction (XRD) patterns of AZO-mica and AZO on PET structures are presented in Figure 1a,b. As shown, the identification of AZO peaks is rather complicated due to the shadowing of the AZO reflection with the stronger signal of mica peaks or stronger PET substrate reflection. Hence, more detailed XRD patterns are shown in Figure 2a,b, where the diffraction peaks for ZnO hexagonal wurtzite structure with space group P63mc are detected. For AZO films on mica, the (100) peak at ~31.77° 2Θ is clearly visible. No other peaks for AZO were detectable. The crystalline structure of Al-doped ZnO films depends strongly on the deposition method and film composition. Concerning the AZO layers obtained by ALD, it was observed [31] that the undoped ZnO was preferentially (002) oriented, whereas the AZO films predominantly had the (100) orientation. The crystal orientations were influenced by the Al-doping. As Al_2_O_3_ was introduced, the (002) peak disappeared and the intensity of the (100) peak became dominant [32]. No traces of additional or impurity peaks were observed even in the logarithmic scale. The observation of only AZO (100) diffraction peaks on muscovite mica substrate suggests that AZO is growing on mica with preferable <100> orientation. The estimated lattice constants were a 3.215 ± 0.017 and c 5.245 ± 0.027. The average crystal grains size was calculated as ~59 nm.

In contrast, the XRD pattern of AZO layers on PET shows mostly reflections from the PET substrates and the weak AZO (100) peak. The estimated average crystalline grain size was ~20 nm. The estimated lattice constants were a 3.210 ± 0.001 and c 5.243 ± 0.003. The (002) and (101) peaks of AZO are not detectable and the AZO (110) peak was very weak, indicating the inferior crystallinity of AZO films deposited on PET at relatively lower substrate temperature. The AZO is growing on PET with preferable <100> orientation. It was also observed that the crystalline structure and preferred crystalline grain growth orientation of AZO on mica depend on the deposition method and process temperature and even allows van der Waals epitaxial growth [33].

The surface morphologies and root mean square (RMS) roughness values of blank mica and PET substrates, as well as AZO-mica and AZO on PET layers as-deposited and after bending of 800 cycles, are shown in Figure 3. Freshly cleaved muscovite mica has a very flat surface substrate as shown in Figure 3a, a prerequisite for the growth of high-quality AZO films. The surface roughness of AZO-mica (Figure 3b) does not change significantly after the bending test (Figure 3c). However, the surface roughness of AZO on PET (Figure 3e) increased almost two-fold, presumably due to the surface deformation during bending (Figure 3f).

### 3.2. Optical Properties

Transmittance spectra of AZO on mica and AZO on PET are shown in Figure 4. The spectra of blank mica and PET substrates are also included for reference. As can be seen, the absorption edges of AZO on mica and AZO on PET are shifted to the longer wavelengths (red shift) in comparison to the blank (reference) substrates. The absorption edge originates from the absorption corresponding to the direct transition of electrons from the valence band to the conduction band [34]. The optical transparency of AZO on mica is higher than those of AZO on PET together with the benefit of improved conductivity, discussed in the next sub-section.

We measured the transmittance spectra of AZO on mica and AZO on PET films after completed bending cycles and found there is no change in optical spectra. The data are presented as a dotted line for AZO on mica film for illustration purposes only.

### 3.3. Electrical Properties and Bending Test

The current vs. voltage (I–V) characteristics of AZO film on mica are presented in Appendix A which confirms Ohmic behavior. Figure 5 shows that AZO on mica exhibits higher conductivity than AZO on PET, assumed to be due to the higher deposition temperature and improved crystallinity of the AZO film [33]. To verify the stability of AZO on mica against mechanical bending, a cycling bending test was performed (see Appendix A) As shown in Figure 5, the sheet resistance of AZO on mica remains almost constant, even after continuous bending of up to 800 times with a bending radius of 1.5 mm. It is supposed the stability is due to the unique layered structure of mica and flexibility of the AZO films. In contrast, the sheet resistance of AZO on PET monotonously increases with the number of bending cycles. As shown in Figure 3f, the RMS data of AFM suggested some deformations (cracks) during the bending test of AZO on PET causing probably breaks in the conductivity channels, i.e., the sheet resistance increases with bending cycles.

### 3.4. AZO on Mica for Assembling PDLC Devices

Based on the above performed structural, optical, and electrical characteristics, AZO on mica was applied for PDLC device assembly. The schematic structure of the device is shown in Figure 6a. An experimental optical set-up (shown in Figure 6b) was assembled to measure the transmittance dependence of PDLC devices as a function of the applied voltage.

In general, the PDLC structure can be reversibly switched between a light-scattering state and a transparent state by applying an external electric field, which results from a match or mismatch of refractive indices between the LC molecules and the polymer matrix (LC with positive dielectric anisotropy) [35,36]. This effect is due to the ability of applied voltage to re-orient the LC molecules inside the droplets in order to match the LC’s refractive index to that of the polymer matrix. When no external applied voltage is applied, due to the mismatch between the refractive indexes of LC molecules and polymer matrix, the randomly oriented LC molecules inside the polymer matrix scatter the incident light and caused PDLC film to appear opaque (“off state” shown in Figure 6c, left side). Under the application of an electric field, LC molecules inside the droplets are able to align towards the electric field direction such that the refractive indices between the polymer and ordinary axis of LC molecules match, resulting in a transparent state (“on state” shown at Figure 6c, right side).

A typical transmittance dependence as a function of the applied voltage of AZO on a mica PDLC device is shown in Figure 7a. The device functionality can be explained as follows: at the initial state (without an applied voltage (*V_off_*)), the LC molecules in the PDLC structure are randomly oriented inside the droplets, which cause a light scattering effect when the light passes through the device. As a result, the opaque state (“off” state) appears. However, when a voltage is applied (*V_on_*), the electric field forces the LC’s nematic director to align along the direction of the electric field, allowing the light to pass through the LC droplets. As a result, the transparent state (“on” state) appears. This way, the transmitted light intensity through the PDLC structure can be controlled by the application of an external voltage.

Here, we defined the threshold voltage, *V_th_*, as an applied voltage value required to reach 10% of the maximum transmittance T (*T*_10%_), e.g., to turn on the PDLC cell. Subsequently, the saturation voltage, *V_sat_*, is the applied voltage value, required to reach 90% of the maximum transmittance T (*T*_90%_). The measured threshold voltage, *V_th_*, of AZO on the mica PDLC device is around 5.8 V, and the saturation voltage, *V_sa_**_t_*, is around 16.6 V (Figure 7a). These values are slightly better than the results presented at [26] for PDLC devices using AZO on PET.

In addition, the response time of the AZO on the mica PDLC device was measured using the set-up shown in Figure 6b. A rectangular pulse with a certain voltage in the working range (between the on and off states of the device) was applied, visualized by a digital oscilloscope. By measuring the transmitted light intensity during the time, a typical dynamic curve is obtained, where the red curve represents the voltage change and the black curve is the transmitted light intensity change (Figure 7b). We denote the response time *τ_r_* as a rise time necessary for the intensity to reach 0.9 (or 90%) of the saturation intensity and the fall time *τ_f_* to drop to 0.1 (or 10%) of the saturation intensity. The time required for transmittance to change from *T*_10%_ to *T*_90%_ is the response time. The obtained response time and the decay time values of ~78 ms and ~74 ms, respectively are in good agreement with those reported for PDLC-based devices using other transparent contacts [37,38,39].

To summarize, the AZO films on mica deposited by the ALD technique support excellent optical transparency, low sheet resistance, and bending stability. The PDLC device performance confirms the feasibility of AZO on mica structure as an alternative flexible transparent conductor.

## 4. Conclusions

In summary, we demonstrated a highly flexible light shutter device using PDLC and AZO on mica transparent conductive films. Structural, optical properties, and the sheet resistance bending durability of AZO on mica were systematically studied and discussed. Flexible PDLC devices were successfully fabricated, and the measured driving voltage and response time values show promising electro-optical functionality. The presented results opened the great potential for integration of AZO transparent conductive layers on mica substrates into the next generation ITO-free flexible optoelectronics.

## Figures and Tables

**Figure 1 nanomaterials-11-01011-f001:**
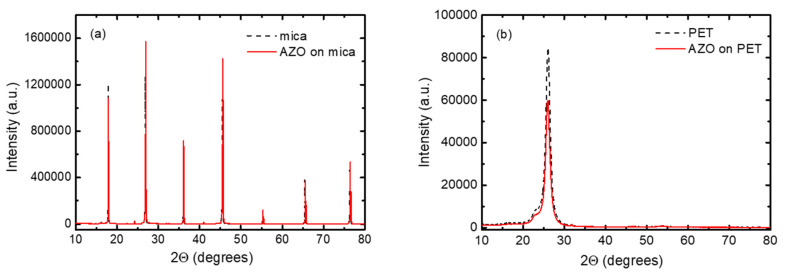
XRD of (**a**) AZO on mica and (**b**) AZO on PET (blank mica and PET are included as references).

**Figure 2 nanomaterials-11-01011-f002:**
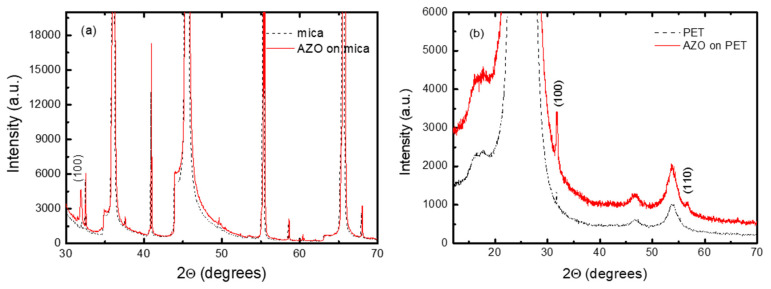
Detailed XRD patterns of (**a**) AZO on mica and (**b**) AZO on PET (blank mica and PET are included as references).

**Figure 3 nanomaterials-11-01011-f003:**
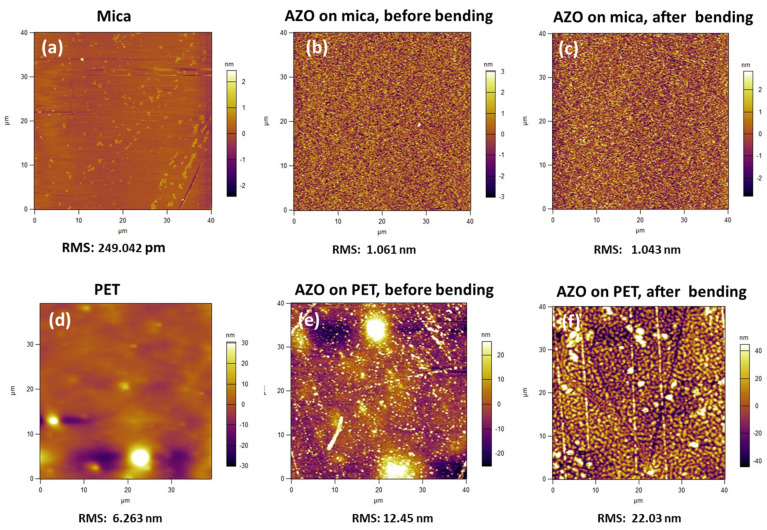
AFM images: (**a**–**c**) AZO on mica (as-deposited and after bending) and blank mica and (**d**–**f**) AZO on PET (as-deposited and after bending test) and blank PET, respectively.

**Figure 4 nanomaterials-11-01011-f004:**
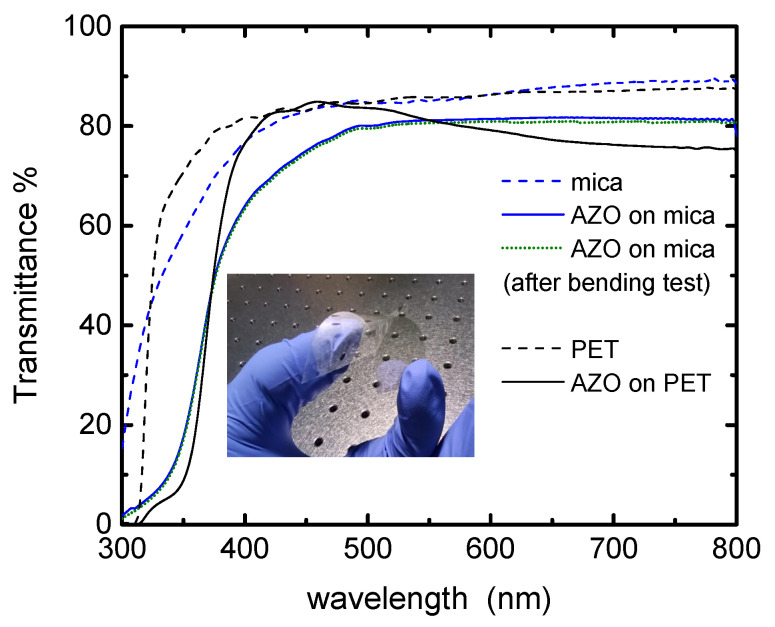
Transmittance spectra of AZO on mica and AZO on PET substrates. Transmittance spectra of mica and PET blank substrates are shown for reference (dash lines), as well as AZO on mica after bending (dot line). Inset image-photograph of AZO on mica substrate.

**Figure 5 nanomaterials-11-01011-f005:**
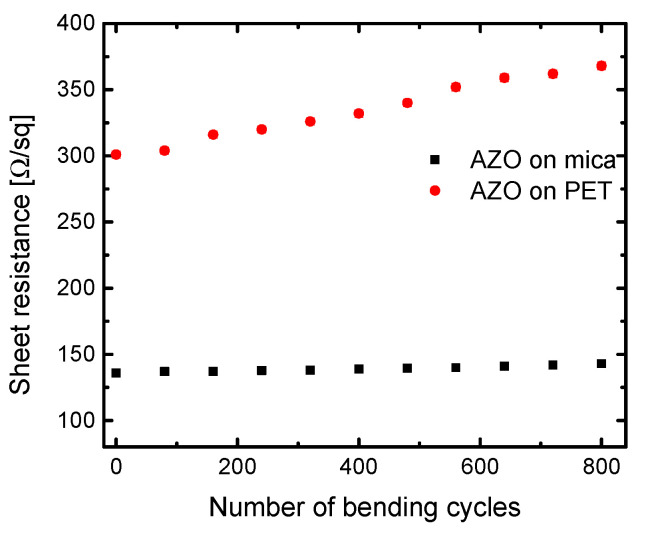
Sheet resistance dependence of the number of bending cycles for AZO on mica and AZO on PET layers.

**Figure 6 nanomaterials-11-01011-f006:**
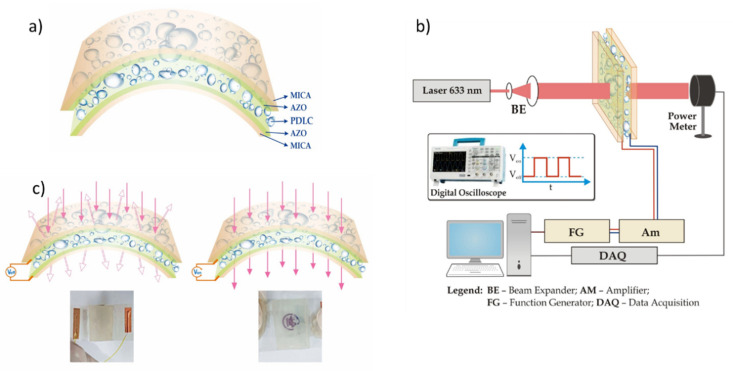
(**a**) Schematic structure of AZO on mica PDLC device (**b**) an experimental set-up to measure voltage-transmittance characteristics and the response time (**c**) principle of work demonstrating image of NCTU of AZO on mica PDLC device at “off” and “on” states. (Appendix A, video demonstration (“off” and “on” states and flexibility)).

**Figure 7 nanomaterials-11-01011-f007:**
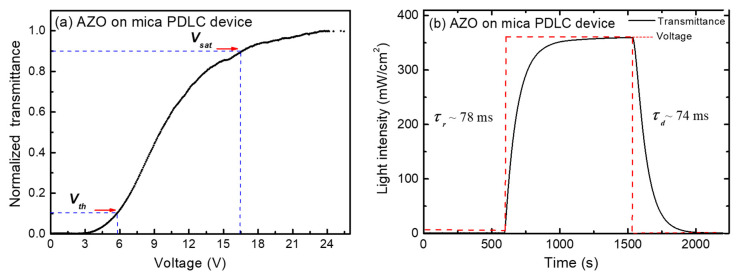
(**a**) Voltage-transmittance curve of AZO on mica PDLC device and (**b**) response time of AZO on mica PDLC device.

## Data Availability

Data is contained within the article or Appendix A.

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
