# Peer review of "ALD Deposited ZnO:Al Films on Mica for Flexible PDLC Devices"

_nanomaterials, 2021, doi:10.3390/nano11041011_

Round 1
Reviewer 1 Report
See enclosed file

Author Response
Nanomaterials, Manuscript ID: nanomaterials-1146981
Title: ALD deposited ZnO: Al films on mica for flexible PDLC devices
by D. Dimitrov et al.
Response to the Reviewers' Comments (1)
We would like to thank the Reviewers for their very useful and appropriate comments. We take their suggestions and advice into consideration during the revision process of our manuscript. Below are the detailed answers and list of changes:
COMMENTS TO THE AUTHOR(S)
This manuscript describes the ALD of Aluminum-doped zinc oxide (AZO) films on mica for flexible PDLC devices. The authors report and compare the properties of AZO films on two flexible substrates (PET-Al2O3 and mica), before and after bending. Eventually, the applicability of AZO as transparent electrode is demonstrated.
The general topic of this manuscript might be of wide interest: ALD-AZO is a well-known attractive material with great potential to replace ITO; while efficient flexible transparent electrodes are becoming a necessity.
This manuscript is interesting but should be somehow improved. In particular, the authors are stating that they compare AZO/Mica and AZO/PET. In reality, they compare two AZO films of different thicknesses, deposited at different temperatures and on different substrates: (mica vs Al2O3, rather than PET). This is very misleading and must be corrected. Also, the film properties and device performance after bending are not enough explored.
We thank for this important note. Indeed, the focus of the manuscript is to study the structural, optical and electro-optical properties of AZO thin films deposited by ALD on mica substrates. We use AZO films on PET only for a brief comparison. As written in the manuscript (first version, Materials and Methods rows 96-99 ) AZO on mica films were deposited at 200°C and AZO films on PET at 100°C . Due to the difference in terms of temperature stability, it is impossible to put both substrates (mica and PET) together at ALD reaction chamber. Indeed, the thicknesses of both films are slightly different (AZO on mica ~ 173 nm) and (AZO on PET~ 150 nm + 15 nm buffer layer) however the difference of 8 nm is considered to be not critical. It was shown that AZO films grown at higher temperatures show better conductivity (T. Dhakal et al. Mater. Res. Soc. Symp. Proc. Vol. 1327, (2011)) therefore we use an optimum temperature of 200 °C for AZO on mica. At temperatures above the glass transition temperature (around 75°C), PET has a compromised ability to retain the form and structure. The deformations of PET surface could protrude to the growing AZO film. Because of this deformation, the film growth and thus its resistance were not uniform. The films grown on the flexible PET substrate required an Al2O3 buffer layer for a uniform and conformal film growth and somewhat lower deposition temperature.
The film properties after bending are studied by AFM, optical transmittance (revised Figure 4 and comments in the revised manuscript) and stability of sheet resistance during bending tests. In term of the device performance, we include some comments at the end of Results and discussion section and also a video demonstration at Appendix (Figure 3A).
Hence, I recommend to ACCEPT it with major revisions.
- The context and the originality of the paper are not clearly stated. There is a wide list of publications, even reviews, on ALD-ZnO that should be cited. Also, status of ALD on flexible substrates should be included.
To the best of our knowledge our research is the first one concerning the ALD of aluminum doped ZnO on mica substrates. We have considered the recommendation and included in the revised version review papers for ALD of ZnO as well as doped ZnO. The status of the ALD on flexible substrates is also updated.
- Reading the features (low cost, higher processing temperatures, stability, transparency, …) reported on mica by the authors, I wonder why people still use other flexible substrates such as PET and PEN. Can you explain?
PET and PEN are organics, unstable with the time and as mentioned above cannot be used at higher deposition temperature, which affect the conductivity. Moreover, fabrication of devices on conventional flexible substrates with superior performance are constrained by the trade-off between processing temperature and device performance. Unlike the conventional substrates, the crystalline structure of mica allows the van der Waals epitaxial growth of oxide thin films resulting in optimal performance. (Ma, C. H.; Lin, J. C.; Liu, H. J.; Do, T. H.; Zhu, Y. M.; Ha, T. D.; Zhan, Q.; Juang, J. Y.; He, Q.; Arenholz, E.; Chiu, P. W.; Chu, Y. H. Van der Waals Epitaxy of Functional MoO2 Film on Mica for Flexible Electronics. Appl. Phys. Lett. 2016, 108, 253104). Flexible mica substrate is proposed as an alternative strategy to circumvent the trade-off issue via the heteroepitaxial growth of transparent conducting oxides (TCO) with performance comparable to that of their rigid counterparts (Bitla et al., ACS Appl. Mater. Interfaces 2016, 8, 47, 32401–32407). Epitaxial TCO thin film was deposited on muscovite mica via pulsed laser deposition using commercial ITO and AZO targets, as well as by RF magnetron sputtering. The deposition process was carried out at substrate temperatures of 400 - 420 °C. In case of AZO layers on mica by sputtering a post deposition annealing at 500 °C is applied to improve electrical conductivity. In our research we are studying the properties and application of AZO films obtained by ALD on mica substrates using lower thermal budget and without post-deposition annealing. It is the first time to use mica as substrate for ALD of AZO films to the best of our knowledge.
- The ALD on flexible substrate can be technically difficult when the substrate is naturally bending in the reaction chamber. If the authors have used any trick, that should be mentioned in the Materials and Methods section
The substrates were fixed with a heat resistant tape (temperature range is specified as -75 to +260°C) in ALD reaction chamber. This is added in Materials and Methods section.
- Why are the authors preventing a possible diffusion possible from PET to AZO? And which? Are such barrier layers common?
Research has shown that organometallic precursors penetrate the subsurface of the polymer and react with functional groups along the backbone of the polymer producing coatings with a graded finish as for example trimethyl aluminum (TMA) infiltration in polyethylene terephthalate (PET) (R. P. Padbury and J. S. Jur, Journal of Vacuum Science & Technology A 33, 01A112 (2015)
Buffer layers are very common in ALD on polymer substrates. Prior to the AZO deposition, a buffer layer usually of Al2O3 is used as an interface to avoid diffusion of chemical species to and from the polymeric substrates, improve adhesion and reduce cracks formation. Al2O3 adhesion to polymer substrates is remarkable due to the covalent bonding (Chang, R. C. et al. Applied mechanics and materials 479–480, 80–85 (2014)).
- Throughout the manuscript, the authors are stating that they compare AZO/Mica and AZO/PET. In reality, they compare two AZO films of different thicknesses, deposited at different temperatures and on different substrates: (mica vs Al2O3, rather than PET). This is very misleading and must be corrected.
We answer to this important point above.
- The fitting procedure of ellipsometric measurements should be included
Ellipsometric measurements for Si reference substrates were performed using a Woollam M2000D rotating compensator spectroscopic ellipsometer with a wavelength range from 193 to 1000 nm. The film thickness and the optical constants were determined by fitting the experimental Ψ and Δ data. The experimental Ψ and Δ were analyzed using a three-layer model consisting of a Si substrate with SiO2 native oxide as a first layer, a ZnO layer as a second layer and a roughness layer as a third layer. For the Si substrate and the native oxide, the data from the database of CompleteEASE Woollam data analysis software was used. The ZnO:Al layer was modeled using a PSemi-M0 and two Gaussian oscillators. The roughness layer for all samples is modeled by Bruggeman’s EMA (Effective Medium Approximation) of 50 % voids and 50 % bulk material (B S Blagoev et al. 2016 J. Phys.: Conf. Ser. 764 012004)
The fitting procedure of ellipsometric measurements is included in the revised version.
- Regarding the optical properties, the lower transmission in the 700-800 nm range could be related to a difference in carrier concentration? Can you discuss and add simple electrical characterization measurement such as Hall effects?
We highly appreciate this important remark. Due to the very limited time for revision and continuous pandemic situation, it is impossible to perform the Hall effects measurements, however, we added some discussion from other references and plan to perform the Hall measurements as next research.
The carrier concentration is strongly correlated with the deposition temperature. The carrier concentration obtained at a high deposition temperature was ~2 orders of magnitude higher than that obtained at a lower temperature. (W. J. Maeng et al. Journal of Vacuum Science & Technology B 30, 031210 (2012)). Therefore, the carrier concentration in AZO on PET films deposited at 100 C is definitely lower than those of AZO on mica films and influenced the optical transmission. Also there is composition dependence of carrier concentrations and structural effects.
- What about the film properties (optical, electrical) and device performance before/after bending?
We thank about this important point, which was included in more details in a revised version.
In case of the film properties (optical, electrical) – The film properties after bending are studied by AFM, optical transmittance (revised Figure 4 and comments in the revised manuscript) and stability of sheet resistance during bending tests. In term of the device performance, we include some comments at the end of Results and discussion section and also a video demonstration at Appendix (Figure 3A).
The device performance before/after bending is also very stable. We add a video demonstration focused on the bending stability at Appendix part, Figure A3.
Once again, we thanks to all Reviewers for very essential and important comments. We consider all their remarks during revision process of our manuscript. We hope the revised version will match the requirements of Nanomaterials journal.

Reviewer 2 Report
The manuscript by Dimitrov et al. (nanomaterials-1146981) demonstrates the transmittance of AZO/mica PDLC device depending on the external voltage as well as the materials characteristics. In spite of interesting performances, the reviewer does not feel to satisfy the publishing criteria in Nanomaterials, because no significant improvement is observed in comparison with previous results that the authors reported (AZO/PET and TiO2 layers). Here are several comments as below, would refer before submitting to other journals.
1) Abbreviation issue; ALD (not use)/PDLC (order)
2) The reviewer thinks that the position of Figure 1(b) should replace into Figure 2(a).
3) The authors would be better to apply logarithmic scale of XRD intensity for more accurate demonstration and analysis of XRD.
4) The significant figures that the authors used are too long. In general, the digit of significant figure has used as 3.
5) The reviewer does not understand that AZO is growing on mica with preferable <100> orientation. From the XRD peak of AZO/mica, the authors claimed (100) peak around 32 deg is only detected. However, many papers on AZO showed the three major peaks around 32, 34, and 36 deg. Even if the peak appearing at 36 deg (101) is buried by the mica, the peak around 34 deg (002) should appear. From the limited XRD peak, how can the authors claim the preferable growth direction.
Author Response
Nanomaterials, Manuscript ID: nanomaterials-1146981
Title: ALD deposited ZnO: Al films on mica for flexible PDLC devices
by D. Dimitrov et al.
Response to the Reviewers' Comments (2)
Reviewer comments:
First of all, we would like to thank the Reviewers for their very useful and appropriate comments. We take their suggestions and advice into consideration during the revision process of our manuscript. Below are the detailed answers and list of changes:
The manuscript by Dimitrov et al. (nanomaterials-1146981) demonstrates the transmittance of AZO/mica PDLC device depending on the external voltage as well as the materials characteristics. In spite of interesting performances, the reviewer does not feel to satisfy the publishing criteria in Nanomaterials, because no significant improvement is observed in comparison with previous results that the authors reported (AZO/PET and TiO2 layers). Here are several comments as below, would refer before submitting to other journals.
The previous published results on three-layer system TiO2-Ag-TiO2 [Ref.38] (not just TiO2 as the Reviewer noted) are focused on numerical simulations that suggest the best match between the deposition time and individual layer thickness. Since the TiO2-Ag-TiO2 were deposited on a PET, as flexible substrate, we demonstrate application of PDLC structure, however the working parameters are completely different than these presented in current manuscript.
In terms of AZO deposited on PET substrate, at the time the results were published [Ref.37] we do not have yet AZO on mica samples for compassion. Therefore, to the best of our knowledge our current research is the first one concerning the ALD of aluminum doped ZnO on mica substrates.
1) Abbreviation issue; ALD (not use)/PDLC (order)
It is not clear what the issue is about. ALD stands for Atomic Layer Deposition, while PDLC for Polymer Dispersed Liquid Crystal
2) The reviewer thinks that the position of Figure 1(b) should replace into Figure 2(a).
corrected
3) The authors would be better to apply logarithmic scale of XRD intensity for more accurate demonstration and analysis of XRD.
Figure. XRD of AZO/mica in logarithmic scale (attached file)
There is no additional or impurities peaks detected when XRD spectra are presented at logarithmic scale. This comment is included in the revised version.
4) The significant figures that the authors used are too long. In general, the digit of significant figure has used as 3.
corrected
5) The reviewer does not understand that AZO is growing on mica with preferable <100> orientation. From the XRD peak of AZO/mica, the authors claimed (100) peak around 32 deg is only detected. However, many papers on AZO showed the three major peaks around 32, 34, and 36 deg. Even if the peak appearing at 36 deg (101) is buried by the mica, the peak around 34 deg (002) should appear. From the limited XRD peak, how can the authors claim the preferable growth direction.
The crystalline structure of Al-doped ZnO films depends strongly of the deposition method. Concerning the AZO layers obtained by ALD, Banerjee et al. (1) demonstrated that the ZnO films grown without any trimethylaluminum (Al(CH3)3) (TMA)-DI water cycle exhibited a polycrystalline state having (100), (002), (101), and (110) orientations. However, as Al2O3 was introduced, the (002) peak disappeared and the intensity of the (100) peak became dominant. Similar results were obtained by Kwon et al. (2) They also found that the crystal orientations were influenced by the Al-doping. They observed that the undoped ZnO was preferentially (002) oriented, whereas the AZO films predominantly had the (100) orientation. In the work carried out by Saarenpaa et al. (3), temperature dependent growth orientation was observed. They observed that at 150 °C the [100] direction was dominant, whereas at 250 °C the [002] direction dominated the crystalline growth. Furthermore, for the Al doped ZnO films in Ref.4, the intensity of (100) peak was dominant among other weaker peaks irrespective of Al-doping percentage, growth temperature, and substrate used.
1.P. Banerjee et al., J. Appl. Phys. 108, 043504 (2010). 2.S.J. Kwon, Jpn. J. Appl. Phys. 44, 1062 (2005). 3.H. Saarenpaa et al., Solar Energy Mater. Solar Cells 94, 1379 (2010) 4. T. Dhakal et al. Journal of Vacuum Science & Technology A 30, 021202 (2012)
This comment is included in the revised version.
Once again, we thanks to all Reviewers for very essential and important comments. We consider all their remarks during revision process of our manuscript. We hope the revised version will match the requirements of Nanomaterials journal.
Reviewer 3 Report
Reviewer’s comments ragarding:
Manuscript details:
Journal: Nanomaterials
Manuscript ID: nanomaterials-1146981
Type of manuscript: Article
Title: ALD deposited ZnO: Al films on mica for flexible PDLC de-vices
Authors: Dimitre Z. Dimitrov *, Vera Marinova *, Chih Yao Ho, Dimitrina
Petrova, Blagovest Napoleonov, Blagoy Blagoev, Velichka Strijkova, Ken Yuh
Hsu, Shiuan Huei Lin, Jenh-Yih Juang
- In Figure 2a, the explanation of the figure bellow it is not correct: “XRD of (a) AZO/PET “. Please correct it.
- In the explanation of figure 3: “Figure 3. AFM images: (a, b, c) AZO/mica (as deposited and after bending) and blank mica and (d, e, f) AZO/PET (as deposited and after bending test) and blank PET, respectively.” : When mentioning (a,b,c), since blank mica is the first (a), it should be explained the first, same four figures d,e,f. Mention the indicative of the figure after each explanation, for example: as deposited (b) and after bending (c).
- Page 7, rows 222-228, the explanation of the PDLC functioning given by the authors is valid only for LC with positive dielectric anisotropy. This should be mentioned.
- The authors should give details about the calculus of the normalized transmission in Fig.7a
- The authors should give details about the experiment and the calculus of response time and the decay time values
My conclusion: Accept after minor revision.
Author Response
Nanomaterials, Manuscript ID: nanomaterials-1146981
Title: ALD deposited ZnO: Al films on mica for flexible PDLC devices
by D. Dimitrov et al.
Response to the Reviewers' Comments (3)
First of all, we would like to thank the Reviewers for their very useful and appropriate comments. We take their suggestions and advice into consideration during the revision process of our manuscript. Below are the detailed answers and list of changes:
- In Figure 2a, the explanation of the figure bellow it is not correct: “XRD of (a) AZO/PET“. Please correct it.
Thank you for this remark. We correct the Figure 2(a) captions
- In the explanation of figure 3: “Figure 3. AFM images: (a, b, c) AZO/mica (as deposited and after bending) and blank mica and (d, e, f) AZO/PET (as deposited and after bending test) and blank PET, respectively.”: When mentioning (a,b,c), since blank mica is the first (a), it should be explained the first, same four figures d,e,f. Mention the indicative of the figure after each explanation, for example: as deposited (b) and after bending (c).
Thank you very much to point out, this is very important and corrected in the revised version.
- Page 7, rows 222-228, the explanation of the PDLC functioning given by the authors is valid only for LC with positive dielectric anisotropy. This should be mentioned.
We add this point
- The authors should give details about the calculus of the normalized transmission in Fig.7a
The transmission behavior at Fig.7a is normalized to unity (arbitrary units). By this presentation, it is easy to define the driving voltage Vth (an applied voltage value required to reach 10% of the maximum transmittance T (T10%)) and the saturation voltage Vsat (the applied voltage value, required to reach 90% of the maximum transmittance T (T90%)).
- The authors should give details about the experiment and the calculus of response time and the decay time values
To measure the response time, we use the set-up shown at Fig. 6 (revised set-up). A rectangular pulse with certain voltage in the working range of the device has been applied (from V=0 to V≥ Vsat, following the obtained values from Fig.7(a)). Characteristic response curves were recorded with a digital storage oscilloscope. We denote the response time τr as a rise time necessary the intensity to reach 0.9 (or 90%) of the saturation intensity and the fall time τf to drop to 0.1 (or 10%) of the saturation intensity. The time required for the device to change transmittance from T10% to T90% is the response time of the device.
Once again, we thanks to all Reviewers for very essential and important comments. We consider all their remarks during revision process of our manuscript. We hope the revised version will match the requirements of Nanomaterials journal.

Round 2
Reviewer 1 Report
Revised version recommanded for acceptance
Reviewer 2 Report
Agree to publish